# The Importance of Selected Aspects of a Company's Reputation for Individual Stock Market Investors—Evidence from Polish Capital Market

**Tomasz L. Nawrocki * and Danuta Szwajca**

Faculty of Organization and Management, Silesian University of Technology, 41-800 Zabrze, Poland; danuta.szwajca@polsl.pl
* Correspondence: tomasz.nawrocki@polsl.pl

**Abstract:** In recent decades, the company's reputation has become an important signal and a decision-making stimulus for one of the key stakeholder groups—investors. Reputation includes both cognitive and affective aspects that investors may be more or less guided by. The article examines the importance of selected aspects of reputation for individual stock market investors on the capital market in Poland. The research used the method of an internet survey addressed to 417 individual investors, and the survey results allowed the answering of five research questions. The research results showed that from the point of view of individual investors operating on the Polish capital market, the informational aspects of companies' reputations are slightly more important than the financial and growth aspects, and the least important are the social aspects, although a considerable internal differentiation of the significance of individual sub-criteria was noted. This study is the first to examine the importance of various aspects of reputation among Polish individual investors and one of the few such studies on an international scale.

**Keywords:** corporate reputation; reputation management; individual investors; investor motivations

## 1. Introduction

Reputation has for several decades been considered one of the most valuable assets of a company, both by theoreticians and practitioners of management. The importance of reputation is demonstrated by several theories that have arisen over this time, as follows: strategy theory [1,2], resource-based value theory [3,4], stakeholder theory [5,6] and signaling theory [7,8].

A good reputation is very beneficial to a company. Over 20 years of research in various countries shows that companies with a strong, positive reputation achieve better financial results [9,10] and higher share prices [11,12]. A good reputation also has an impact on building the company's market value [13,14] and allows companies to survive periods of crisis or worse economic conditions [15–17]. All these benefits, among others, result from the fact that reputation becomes one of the main factors/determinants of the behavior and decisions of many stakeholder groups [14,18,19]. According to Weber Shandwick's research from 2019, a good reputation has the following benefits: builds and strengthens customer loyalty, improves relationships with suppliers and business partners, reduces employee retention and attracts talent, provides higher stock prices, provides greater support from control institutions and regulators, allows for gaining favor of media and journalists, attracts and retains investors, minimizes risk and protects against the negative effects of a crisis [20].

Investors and shareholders are one of the key stakeholder groups. Their decisions about investing capital largely determine the development opportunities and financial results of the enterprise. Therefore, it is very important to identify the reasons and motivations for their investment decisions. The authors have shown in many studies that

reputation is becoming an increasingly important selection criterion for capital donors—individual shareholders, public investors, and investment funds [21–25]. The research conducted by FTI Consulting in 2019 showed that the investors' response to corporate crises is determined more by reputation than by financial performance, that is, investors are "driven more by reputation than by numbers" [26].

Reputation is a multidimensional, multi-faceted, complex category that includes cognitive and affective aspects that may be of varying importance to investors in the decision-making process. Until now, many authors and researchers have discussed the importance of reputation for investors in their studies [21–25,27,28], but there is a lack of in-depth research on the role of its various aspects as motives for investor behavior and decisions in the capital market. Therefore, the aim of the article was to fill this gap by examining the importance of selected aspects of reputation, such as the following: informational, financial and growth, as well as social, for this group of business stakeholders. This goal was achieved by conducting an online survey targeted at stock market investors operating on the Polish capital market.

The obtained survey results allowed for the answering of the following research questions:

Q1  How important is corporate reputation for investors?
Q2  Which aspects of corporate reputation—informational, financial and growth or social—are the most important for investors?
Q3  Which determinants of the informational aspect of corporate reputation are the most important for investors?
Q4  Which determinants of the financial and growth aspect of corporate reputation are the most important for investors?
Q5  Which determinants of the social aspect of corporate reputation are the most important for investors?
Q6  What time horizon, according to individual investors, should be taken into account when assessing corporate reputation?

The structure of the article, which consists of the following sections, follows the introduction. Section 2 presents the essence and multidimensional nature of reputation and the non-financial aspects of reputation as a motive for investor decisions and behavior. Section 3 shows the research methodology and research questions. Section 4 presents the results of the questionnaire survey. Section 5 provides a discussion and Section 6 presents the conclusions.

## 2. Literature Review

### 2.1. Reputation as a Multidimensional Construct

Reputation is a very complex, interdisciplinary and difficult-to-define category, which is emphasized by many authors [29–33]. Therefore, reputation is the subject of multi-faceted research and analysis conducted by specialists in various fields, such as economics, management, finance, marketing, sociology, psychology and ethics [34–38]. The definition of reputation formulated by Fombrun [39] is considered to be the leading one, according to which reputation is a general assessment of the organization's activities to date and predictions about its future, formulated by its external and internal stakeholders. This definition defines reputation as an overall assessment of a company's performance, based on the collective, aggregated perception of various stakeholder groups. Each of these groups has different relations with the company and different expectations towards it; therefore, each evaluates it from a different perspective, through the prism of their own needs, interests and ideas. Customers evaluate the company through the prism of the quality of products and the service as well as prices; employees evaluate on the basis of the attractiveness of working conditions and treatment; investors prioritize financial results, the profit rate and the development potential of the company; co-operators assess the reliability and profitability of cooperation; and important factors for local communities are the responsibility and social commitment of the company. Therefore, the company

does not have one, but many reputations (the company's reputation as a supplier, investor, employer, business partner, socially responsible organization, etc.), and each of them may be different [40,41]. When building its reputation, a company must take into account the points of view of these groups and their often contradictory expectations [42–44].

It is widely recognized that one of the key stakeholder groups are investors and shareholders, especially in the case of listed companies. According to the results of the research by Weber Shandwick [20], global executives consider that the most important element to a company's reputation is the perception of the following three groups of stakeholders: customers, investors and employees (Table 1).

**Table 1.** Importance of stakeholder perceptions to company reputation.

| Stakeholder Group | Very/Somewhat Important [%] |
|---|---|
| Customers | 87 |
| Investors | 86 |
| Employees | 83 |
| Suppliers and partners | 80 |
| People in the local community | 75 |
| Government officials and regulators | 74 |
| The media | 73 |
| People and social media | 68 |
| Non-profits, advocacy groups or non-governmental organizations | 66 |

Source: own work based on The State of Corporate Reputation in 2020: Everything Matters Now. Weber Shandwick, 1.14.2020, p. 9. https://www.webershandwick.com/wp-content/uploads/2020/01/The-State-of-Corporate-Reputation-in-2020_executive-summary_FINAL.pdf (accessed on 15 October 2021).

Stakeholders evaluate the company based on rational premises, which are based on the cognitive sphere, as well as emotional premises, based on the affective sphere. Therefore, the two basic dimensions of reputation are indicated as the following: cognitive and affective. In his concept of reputation measurement, Schwaiger [45] indicated that reputation should be treated as a two-dimensional construct, including the assessment of the company's competencies and the assessment of sympathy towards it. Each of these dimensions has been operationalized by the following three indicators [46]:

- Competence items: (1) [The company] is a top competitor in its market, (2) As far as I know, [the company] is respected worldwide, and (3) I believe that [the company] performs at a premium level;
- Likeability items: (1) [The company] is a company that I can better identify with than with other companies, (2) [The company] is a company that I would miss more than other companies if it did not exist anymore, and (3) I regard [the company] as a likeable company.

Lange, Lee and Dai [47] in the developed model of reputation indicated two of the essential dimensions of a company's reputation, as follows: being known for something and generalized favorability. The dimension of *being known for something* concerns the cognitive sphere, is based on "hard", rational elements, and refers to the level of professionalism, competence and professionalism assessed by particular groups of stakeholders. On the other hand, the dimension of *generalized favorability* relates to the affective sphere, concerns "soft", emotional aspects, and concerns the assessment of the company in terms of its honesty, compliance with the law, ethical standards and respecting the values valued by stakeholders. These dimensions are referred to as competence reputation and character reputation [48].

With regard to investors, the cognitive aspects are based on hard financial data, while the affective aspects of the emotional sphere are based on non-financial information about the company. The non-financial aspects of reputation driven by investors, and the investors' sympathy towards the company, may result from subjective premises (perception of the reporting method, communication method), but also may result from the investors' sus-

ceptibility to general trends and socio-political changes (increased power and importance of stakeholders, popularization of CSR and the principles of sustainable development, expectations regarding transparency and the transparency of enterprises' activities).

*2.2. Different Dimensions of Reputation in the Context of Investor Decision Motives and Behavior Reputation as a Multidimensional Construct*

Investors, in their decisions to locate capital, are guided by many different factors, both rational and emotional. The behaviors and motivations of stock market investors have been the subject of research by economists and financial analysts for many years, who formulate appropriate theories and decision models [49,50]. In early decision-making theories, it was assumed that investors were guided mainly by rational premises, based on "hard" facts, as follows: financial results and reliable information about the company's development prospects. Therefore, the conducted research focused on these factors [51–53]. The development of behavioral economics [54], the emergence of a new field of finance called "behavioural finance" and subsequent research have shown that investors are also driven by psychological (behavioral) and social factors [49,55–59], and the decision-making process itself is much more complex [60]. It has been found that psychological and social factors have an indirect effect, first influencing the formation of a company's reputation, and then the reputation influencing investor decisions [28].

The research conducted in recent years in many countries shows the growing importance of the non-financial, affective aspects of reputation and their impact on investor decisions and the financial performance of companies [61–64]. Based on the literature review, the following aspects can be identified [49,55–57,59,65]:

- The company's involvement in CSR (Corporate Social Responsibility) activities;
- The manner and style of reporting information;
- The opinions of other stakeholder groups and the company's approach to other stakeholders;
- The transparency and communication.

The involvement of enterprises in solving social and environmental problems and their compliance with ethical standards are becoming more and more important as decision-making motives for many stakeholder groups, including investors [66–68]. The positive impact of a company's involvement in CSR on its reputation and the perception of the company by investors, and thus on their investment decisions, is demonstrated by many authors in their research [19,69,70].

Investors need a large amount of information about the company's activities that is not only financial. In recent years, the demand for non-financial data has been growing; therefore the content of reports that contain more and more non-financial information has changed [71,72]. For investors, the content, quantity and credibility of information are not only important, but also the quality of reporting [73] and the manner, style and tone of information transmission [74,75].

Non-financial motivations of investors also include a kind of empathy, i.e., taking into account the opinions about the company of other stakeholder groups, not only regulators or rating agencies, which is obvious. It is mainly about customer opinions and satisfaction [76,77], the positive attitude of the society and public opinion [46] and other stakeholders [78]. According to the Edelman Trust Barometer [79], 84% of institutional investors believe that maximizing shareholder returns can no longer be a corporation's prime goal and that the interests of shareholders should be balanced with the interests of employees, customers, suppliers and local communities.

Many authors emphasize the importance of transparency and the credibility of the message, which builds trust in the company, and trust is the foundation of reputation [80,81]. Analyzes and research by other authors indicate the large role of the quality and reliability of communication in building investor relations and creating a positive image and reputation of the company [82,83].

A good reputation is also a kind of buffer protecting the company in the event of economic crises [84,85], as well as internal crises [15,86]. Investors during crises are more influenced by reputation than by the company's current financial results in their decisions [26].

Our analysis of the literature allows us to answer the first research question (Q1). In recent years, the company's reputation has become an increasingly important motive for investor behavior and investment decisions. This is due to the fact that reputation is one of the most valuable resources of a modern enterprise. Firms with a strong, positive reputation achieve better financial results, higher stock prices and rates of return, compete more effectively in the marketplace, and gain a long-term advantage. As a result, investing in reputable companies involves less risk.

## 3. Materials and Methods

The main goal of the article, i.e., to examine the importance of selected aspects of companies' reputations for individual investors, including informational, financial and growth aspects as well as social issues, was achieved by conducting a survey using the Google Docs form. The link to the survey form was available on one of the Polish stock exchange portals (StockWatch.pl) and its completion was voluntary and anonymous. The questionnaires were completed by 417 respondents (individual investors) out of 1.356 million (number of brokerage accounts in Poland at the end of November 2021). It allowed the main goal to be achieved and answered the research questions assuming a confidence level of 0.95, a maximum error of 5% and a fraction of 0.5. The detailed metrics of the study participants, taking into account their structure by gender, age and experience, are presented in Table 2. For comparison, Table 2 also shows the structure of respondents in periodic surveys of individual investors in Poland carried out by the Association of Individual Investors in Poland.

**Table 2.** Sample characterization.

| Characteristics of Survey Participants | Research | | Research of Association of Individual Investors (Poland) |
| --- | --- | --- | --- |
| *Gender* | | | |
| Male | 316 | 75.8% | 90.2% |
| Female | 101 | 24.2% | 9.8% |
| *Age* | | | |
| Less than 25 years (<25) | 161 | 38.6% | 8.0% |
| 25–45 years (25–45) | 180 | 43.2% | 60.5% |
| Above 45 years (45<) | 76 | 18.2% | 31.5% |
| *Investment experience* | | | |
| Less than 1 year (<1) | 176 | 42.2% | 47.1% |
| 1–5 years (1–5) | 71 | 17.0% | |
| 5–10 years (5–10) | 53 | 12.7% | 18.8% |
| Above 10 years (10<) | 117 | 28.1% | 34.1% |

Source: own work and https://www.sii.org.pl/14446/aktualnosci/badania-i-rankingi/wyniki-ogolnopolskiego-badania-inwestorow-2021.html (accessed on 10 March 2022).

It is worth noting that the structure of the surveyed investors, due to the distinguished characteristics, is similar in proportions to the structure of individual investors surveyed in periodic surveys conducted by the Association of Individual Investors in Poland.

The survey questionnaire in its main part contained 26 questions about various corporate reputation assessment criteria from the investor's viewpoint. Questions 1 to 7 concerned the informational aspects, questions 8 to 18 the financial and growth aspects and questions 19–26 the social aspects. The respondents evaluated individual criteria of corporate reputation within these three rated areas with a six-point scale from 0 to 5, where 0 means completely irrelevant and 5 very important. By using a six-point scale, the intention was to force respondents to opt for the importance or unimportance of a given reputation criterion and thus eliminate the middle (neutral, indifferent) option of perceiving a given criterion.

The criteria for assessing the reputation of companies included in the survey form were proposed on the basis of literature studies in the field of corporate reputation assessment [24,41,45,64,87], the scope of publicly available information about the situation of listed companies (including, in particular, periodic reports and corporate websites) and the authors' many years of experience in the area of investments on the stock market and fundamental analysis of listed companies (including StockWatch.pl). Firstly, in the course of the literature analysis, different approaches to assessing the reputation of enterprises were identified. In this regard, it can be noted that Helm [41] and Fombrun et al. [87] indicate the difference in interests of various groups of stakeholders shaping the company's reputation, which suggests the need to take into account various criteria and aspects when assessing and measuring it. In turn, Naveed and others [24] prove empirically that financial and non-financial information is essential in building reputation, which, in turn, influences investor decisions. On the other hand, Marzouk [64] and Schwaiger [45] indicate the cognitive and affective dimensions of reputation, which may influence investors' behavior in various ways. Next, the companies' periodic reports and information disclosed on their corporate websites were reviewed to identify the practical use of the various reputation assessment approaches proposed in the literature. In addition to traditional financial data, much attention was also paid to qualitative issues (the way of presenting the information provided, comments on published financial data and descriptive information, including information on corporate social responsibility—CSR— and research and development—R&D—activities). Finally, in the third step, based on the previous investment experience and consultations with other investors and stock market analysts, the theoretical dimension (literature) was compared with the real dimension (periodic reports and corporate websites), which allowed for the development of the final list of proposed criteria for reputation assessment. All survey questions are shown in Table 3.

**Table 3.** List of survey questions (*How do you rate the importance...?*).

| No. | Criterion |
| --- | --- |
| | *Informational aspects* |
| 1 | Transparency of financial information (completeness of items in the financial statements, e.g., reporting of net result on sales, etc.) |
| 2 | Credibility of financial information (e.g., consistency of the data for the reference period with previously published data) |
| 3 | Form of presentation of descriptive information (e.g., photocopy scan, Word, PDF of good quality, etc.) |
| 4 | Extensiveness of descriptive information (comments on results, development plans, innovation potential, R&D and innovation results, human capital and technology) |
| 5 | Disclosure of financial performance forecasts |
| 6 | Meeting financial performance forecasts |
| 7 | Current information about what is going on in the company (e.g., publication of monthly reports, reports on new contracts, etc.) |
| | *Financial and growth aspects* |
| 8 | Financial performance over the past few years (dynamics at individual performance levels, the quality of net profit confirmed by operating cash flows, etc.) |
| 9 | Variability of favorable financial results over the past several years (whether there is an upward trend or whether there is a downward or unstable trend, i.e., an increase at one time and a decrease at another) |
| 10 | Efficiency of operations over the past few years (profitability, work efficiency, etc.) |
| 11 | Maintain positive performance trends/levels in efficiency of operations |

**Table 3.** *Cont.*

| No. | Criterion |
|-----|-----------|
| 12 | Financial condition (short-term and long-term solvency) over the past several years (financial liquidity, debt level and debt servicing capacity, etc.) |
| 13 | Stability over time of a secure financial position (solvency) |
| 14 | Level of innovation potential (patents, licenses, qualified staff and employees, modern fixed assets, access to finance for research and development as well as innovation implementation, etc.) |
| 15 | Development of innovation potential (expenditure on patents, licenses, research and development, employee training, etc.) |
| 16 | Results of innovative activities (implementation of new product, process, marketing or organizational solutions) |
| 17 | Amount of dividends paid (dividend yield) |
| 18 | Regularity of dividends paid |
| *Social aspects* | |
| 19 | Shareholders' structure |
| 20 | Policy of majority shareholders towards minority shareholders |
| 21 | Credibility of the company's management (does what the board says/announces correspond to reality, i.e., is it realized?) |
| 22 | Court cases |
| 23 | Penalties and fines |
| 24 | Honors and awards |
| 25 | Press releases about the company and opinions on web portals |
| 26 | Company's involvement in socially responsible activities (activities concerning local communities, customers, employees, environment, public authorities, investors) |

Source: own work.

In addition to the imposed criteria for assessing reputation within the three areas mentioned above, respondents were also given the opportunity, in the form of an open question, to supplement the list with other, in their view, relevant criteria.

Moreover, the questionnaire asked about what time range in years the evaluation of particular criteria of companies' reputations should have. In this case, an open question was also used.

## 4. Results

Taking into account the assumptions made for the presentation of the research results on the importance of selected aspects of company reputation for individual investors, Table 4 presents the percentage distribution of the obtained evaluation of the importance of particular criteria for reputation evaluation within three considered areas—informational aspects, financial and growth aspects and social aspects.

**Table 4.** Percentage distribution of the received ratings of importance of individual reputation criteria within the three analyzed areas.

| Rated Area | No. | Criterion | 5 | 4 | 3 | 2 | 1 | 0 | Average Importance of Individual Criterion |
|------------|-----|-----------|---|---|---|---|---|---|--------------------------------------------|
| Informational aspects | 1 | Transparency of financial information | 50% | 37% | 9% | 3% | 1% | 0% | 4.33 |
| | 2 | Credibility of financial information | 66% | 27% | 5% | 1% | 0% | 0% | 4.59 |
| | 3 | Form of presentation of descriptive inform . . . | 19% | 31% | 28% | 14% | 5% | 2% | 3.39 |
| | 4 | Extensiveness of descriptive information | 30% | 29% | 26% | 11% | 3% | 1% | 3.68 |
| | 5 | Disclosure of financial performance forecasts | 18% | 28% | 26% | 17% | 8% | 4% | 3.18 |
| | 6 | Meeting financial performance forecasts | 38% | 34% | 17% | 8% | 2% | 2% | 3.93 |
| | 7 | Current information about what is going... | 46% | 30% | 15% | 5% | 3% | 1% | 4.08 |

**Table 4.** *Cont.*

| Rated Area | No. | Criterion | 5 | 4 | 3 | 2 | 1 | 0 | Average Importance of Individual Criterion |
|---|---|---|---|---|---|---|---|---|---|
| Financial and growth aspects | 8 | Financial performance over the past few . . . | 40% | 33% | 21% | 5% | 1% | 0% | 4.08 |
| | 9 | Variability/recurrence of favorable fin . . . | 35% | 38% | 20% | 6% | 1% | 0% | 3.99 |
| | 10 | Efficiency of operations over the past few . . . | 34% | 47% | 15% | 4% | 1% | 0% | 4.08 |
| | 11 | Maintain positive performance trends . . . | 31% | 43% | 21% | 4% | 1% | 0% | 3.99 |
| | 12 | Financial condition (short and long-term . . . | 42% | 37% | 14% | 6% | 1% | 0% | 4.12 |
| | 13 | Stability over time of a secure financial . . . | 40% | 33% | 21% | 6% | 1% | 0% | 4.03 |
| | 14 | Level of innovation potential | 30% | 35% | 23% | 8% | 3% | 0% | 3.81 |
| | 15 | Development of innovation potential | 26% | 37% | 24% | 8% | 4% | 1% | 3.71 |
| | 16 | Results of innovative activities | 30% | 35% | 23% | 8% | 4% | 1% | 3.78 |
| | 17 | Amount of dividends paid (dividend yield | 19% | 28% | 28% | 13% | 8% | 4% | 3.24 |
| | 18 | Regularity of dividends paid | 29% | 24% | 23% | 12% | 9% | 4% | 3.42 |
| Social aspects | 19 | Shareholders' structure | 12% | 30% | 35% | 15% | 5% | 3% | 3.18 |
| | 20 | Policy of majority shareholders towards . . . | 28% | 29% | 27% | 10% | 5% | 2% | 3.61 |
| | 21 | Credibility of the company's management | 64% | 23% | 9% | 3% | 1% | 0% | 4.44 |
| | 22 | Court cases | 21% | 29% | 30% | 13% | 5% | 1% | 3.46 |
| | 23 | Penalties and fines | 24% | 32% | 26% | 12% | 5% | 1% | 3.56 |
| | 24 | Honors and awards | 10% | 18% | 29% | 18% | 15% | 10% | 2.61 |
| | 25 | Press releases about the company and op . . . | 17% | 23% | 30% | 15% | 11% | 4% | 3.08 |
| | 26 | Company's involvement in socially res . . . | 11% | 16% | 21% | 21% | 17% | 13% | 2.44 |

Source: own work.

In the context of research questions 3–5, taking into account the obtained results, it can be stated that from the viewpoint of stock market individual investors, by far the most significant criteria of companies' reputation assessment are *Credibility of financial information* (4.59) and *Transparency of financial information* (4.33) in the informational aspects area (research question 3), as well as *Credibility of the company's management* (4.44) in the social aspects area (research question 5). In the financial and growth area, the highest mean indications of importance were obtained for the following criteria: *Financial condition (short-term and long-term solvency) over the past several years* (4.12), *Financial performance over the past few years* and *Efficiency of operations over the past few years* (both 4.08) (research question 5).

At the same time, the greatest dispersion of respondents' indications as to the relevance of the criteria for assessing the reputation of companies concerned the following: *Company's involvement in socially responsible activities*, *Honours and awards*, and *Press releases about the company and opinions on web portals* in the social aspects area, *Regularity* and *Amount of dividend paid* in the financial and growth aspects area, as well as *Disclosure of financial performance forecasts* in the information aspects area. It should be noted that the above criteria were also characterized by the lowest average indications of importance in their respective subject areas.

The summary of the general research results on the significance of the considered criteria for assessing the reputation of companies from the viewpoint of stock market individual investors is presented in Figure 1, where the average significance ratings for individual criteria are compared with their average rating for a given area (average of the average).

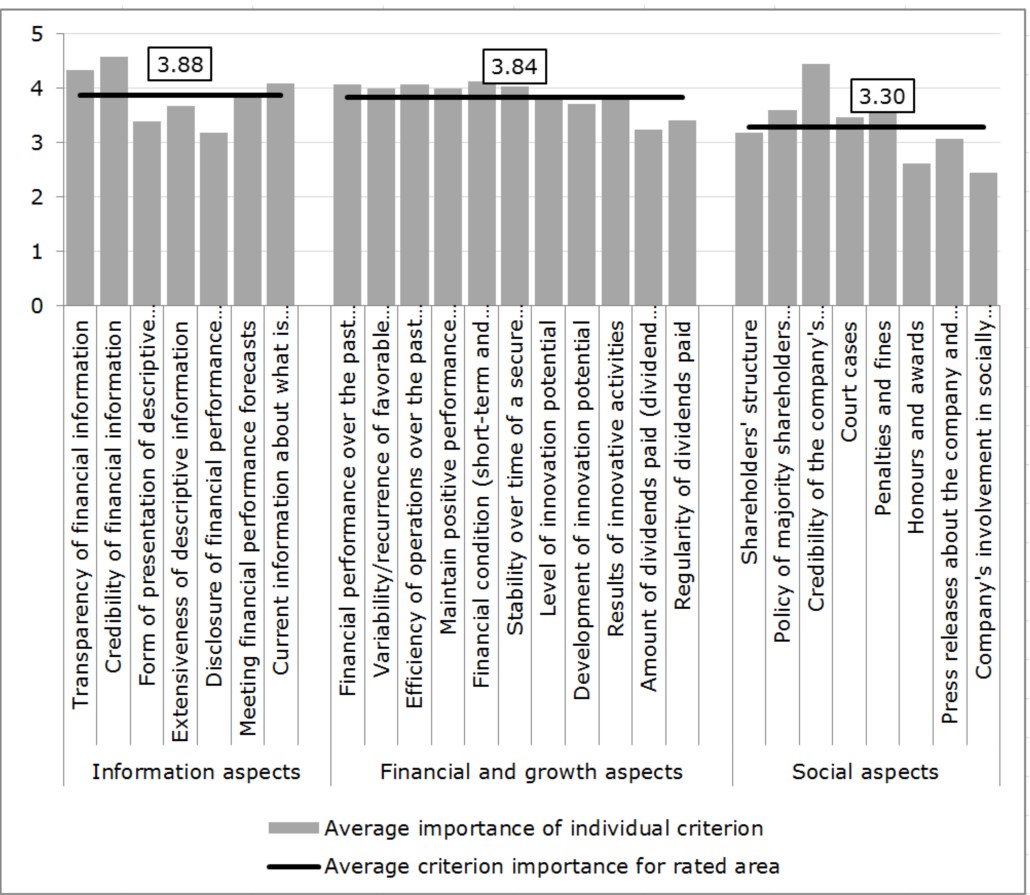

**Figure 1.** The average importance ratings for individual corporate reputation criteria and average ratings for a given area. Source: own work.

In the context of research question 2, the presented results indicate that from the viewpoint of individual investors operating on the Polish capital market, the informational aspects of companies' reputation assessment are even slightly more important than the financial and growth aspects (average 3.88 vs. 3.84). As a group, the social aspects are of the least importance for investors (3.30), although there is quite a significant internal differentiation in the significance of individual partial criteria in it.

As for other criteria for assessing the reputation of companies, which, according to investors, should be taken into account, in response to open-ended questions included in the questionnaire form, 60 respondents indicated suggestions regarding informational aspects, 24 financial and growth aspects and 23 social aspects.

With regard to the informational aspects, the issues of good communication between the company and investors were most often mentioned, including the organization of periodic meetings of management with investors and the presentation of periodic financial results, communicating information in comprehensible language, providing information in a permanent, constant form and access to archival information.

With regard to the financial and growth aspects, *the issues related to investment plans and their scope* were most often mentioned. On the other hand, with regard to the social aspects, the respondents' interest mainly concerned more specific issues in the area of CSR, i.e., *charity activities and sports sponsorship*, *relations with employees, customers and suppliers*, and *involvement in environmental protection*.

The last issue examined during the survey concerned the time range in the years of assessing individual corporate reputation criteria in order to reflect the long-term nature of creating/building this category. Out of 417 completed questionnaires, 254 respondents

(60.9% of the research sample size) answered the open question. The distribution of the received responses is shown in Figure 2.

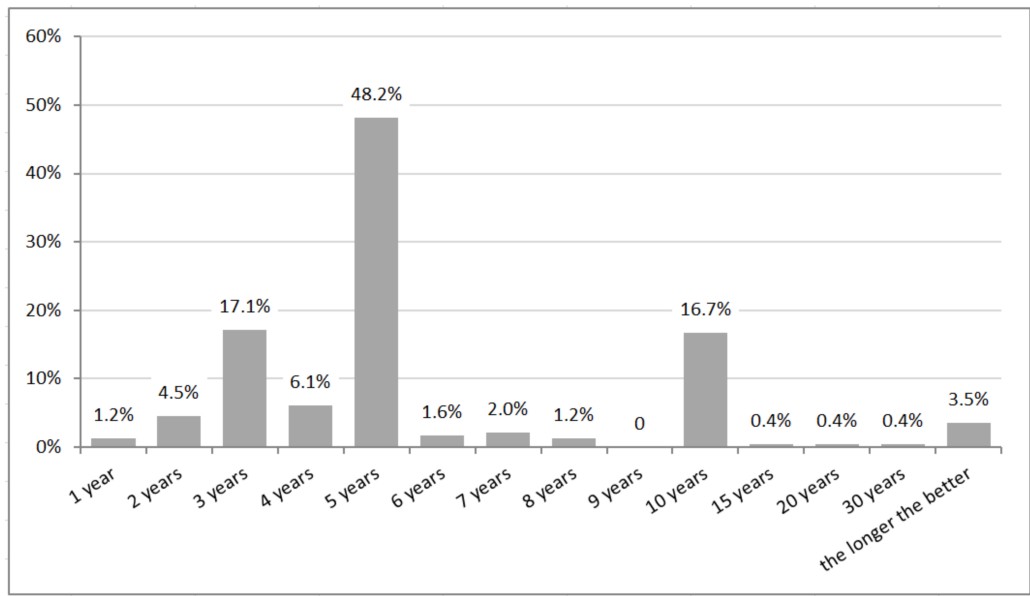

**Figure 2.** The time horizon, which according to individual investors, should be taken into account when assessing corporate reputation. Source: own work.

Taking into account the results obtained in the context of research question 6, it can be concluded that slightly above 70% of respondents' indications concern a period of at least 5 years, of which the highest percentage of indications (48.2%) in the entire research relates to a period of exactly 5 years.

## 5. Discussion

The analysis of the literature and the results of the questionnaire research carried out in the article made it possible to answer the six research questions posed in the introduction. The answer to the first research question (Q1) regarding the importance of reputation for investors was formulated on the basis of the literature analysis carried out in Section 2. This analysis showed that empirical research and concepts formulated in the last several decades have proven the growing importance of non-financial factors (psychological and social), guided by investors in their decisions. One of such factors is the company's reputation and its importance for investors' decisions is revealed especially in the conditions of financial crises or weaker economic conditions.

The next research questions (Q2–Q6) were related to the conducted survey research. The obtained research results showed that the informational aspects of the company's reputation are the most important for individual investors operating on the Polish financial market (Q2). In particular, the *Credibility of financial information* and *Transparency of financial information* (Q3) are important. This is understandable as investors need reliable financial information to optimize their capital allocation decisions and reduce investment risk. This is confirmed by the results of other studies. For example, Blajer-Gołębiewska & Kos [88] prove that investors are more sensitive to information about financial results than to the ethical aspects of reputation. It is also not surprising that the financial and growth aspects are as important as the informative. With regard to this aspect, almost all of the detailed criteria listed (8–16, Table 4) were assessed at the same level of importance. The criteria *Amount of dividends paid* (dividend yield) and *Regularity of dividends paid* (Q4) received slightly lower scores. On the other hand, the relatively low importance of social aspects may be somewhat puzzling, especially in the context of the current trends in the increase in social involvement of companies and the positive attitude of investors to such activities of

the enterprise, as demonstrated by some studies [67,89,90]. Of the eight detailed criteria, the most important were the following: *Credibility of the company's management* and *Policy of majority shareholders towards minority shareholders* (Q5). The lowest rated in importance was *Company's involvement in socially responsible activities* (CSR). To explain such an attitude of Polish investors, several possible, presumptive justifications can be indicated. Firstly, it may result from a rather superficial and stereotypical understanding of the company's social commitment and CSR activities, which in Poland is usually associated mainly with charity and even sponsorship, and this is considered a manifestation of a specific financial mismanagement of the enterprise [91,92]. Secondly, investors expect specific, measurable information about the company's social commitment, while the data of companies on this subject are often too general, descriptive and enigmatic, which results from the lack of uniform CSR reporting standards and the lack of reporting obligations [93]. Thirdly, investors may be quite skeptical about the credibility and declared motives of pro-social activities of the company because often the main intention of these activities is to create a positive image of the company as socially responsible, and the activities themselves are superficial, façade, ad hoc and do not have a strategic basis. Greenwashing practices used for the same purpose, consisting of the transmission of manipulated or even false information about the social activities of the enterprise, are also quite common [94,95]. Moreover, the research conducted on the Polish market is of a pioneering nature and it is difficult to determine on the basis of their results whether the social aspects of reputation were more or less important for investors in the past. The last research question (Q6) concerned the time range of the assessment of the distinguished aspects of reputation. The vast majority of the surveyed investors (75%) indicated a period of at least five years, which seems most understandable due to the fact that reputation is a long-term category, i.e., it takes many years to build it.

## 6. Conclusions

The motives behind the decisions and behavior of stock exchange investors have been the subject of research by economists and financial analysts for many years. In the initial theories and decision models it was assumed that investors were guided mainly by rational premises, based on the assessment of "hard", financial and economic data. The development of behavioral economics and a new field of finance—behavioral finance, as well as empirical research conducted have shown that investors are also guided by non-financial, psychological and social factors. One of such factors is the company's reputation—a multi-faceted category, built on both cognitive and emotional premises. The main aim of the article was to determine the importance of selected aspects of reputation (informational, financial and growth as well as social) for individual investors operating on the Polish financial market. It was achieved through a survey on a sample of 417 investors, assuming a confidence level of 0.95, a maximum error of 5% and a fraction of 0.5.

Due to the considerable length of the detailed results presentation, taking into account the division of the research sample by age, gender and investment experience, this article focuses on the presentation of the results in general terms (without the above-mentioned breakdowns). Taking into account the structure of the research sample (Table 2), it can be said that these results mainly reflect the opinion of male investors (75.8%), aged up to 45 (81.8%) and with investment experience of up to 5 years (59.2%). More detailed research results, taking into account the division of the research sample by gender, age and investment experience, will be presented in separate publications focusing on the indicated thematic areas (informational, financial and growth, social).

The obtained results showed that the most important aspects for the surveyed investors are informational aspects, including, in particular, *Credibility of financial information* (4.59) and *Transparency of financial information* (4.33). The financial and growth aspects are almost as important, with the highest scores for the partial criteria as follows: *Financial condition (short-term and long-term solvency) over the past several years* (4.12), *Financial performance over the past few years* (4.08) and *Efficiency of operations over the past few years* (4.08). The

social aspects described by eight sub-criteria turned out to be the least important for the surveyed investors. Interestingly, the assessments of these criteria showed the greatest internal differentiation. The highest indicators were achieved by: *Credibility of the company's management* (4.44) and *Policy of majority shareholders towards minority shareholders* (3.61), and the lowest were observed for: *Company's involvement in socially responsible activities* (CSR) and *Honors and awards* (2.44).

The research results presented in the article indicate the most important aspects of reputation that are followed by individual investors when making decisions about investing their funds in stocks of listed companies. This knowledge is extremely valuable for company managers, as it allows them to manage corporate reputation more effectively by identifying its critical aspects that should be strengthened and improved. Consequently, it can lead to a more efficient allocation and use of enterprise resources. It is particularly important considering the fact that investors are a priority group of stakeholders of listed companies.

The obtained results allow also for better exploration and understanding of the decision-making mechanisms of individual stock market investors; therefore, they may also be useful to researchers and analysts of capital markets. Thus, they can deepen their knowledge and experience, which will allow them to build more accurate forecasts and predictions about the development of stock prices and other stock market indicators.

Although the research has certain limitations (only the opinions of individual investors on the Polish capital market were examined), it may provide a point of reference for comparisons with the results of similar research conducted in other countries. It may also serve as an introduction to more in-depth research into individual investors' perceptions of the importance of different criteria for assessing companies' reputations within informational, financial and growth as well as social aspects, taking into account different criteria differentiating investors according to factors such as gender, age or investment experience.

**Author Contributions:** Conceptualization, T.L.N. and D.S.; methodology, T.L.N. and D.S.; software, T.L.N. and D.S.; validation, T.L.N. and D.S.; formal analysis, T.L.N. and D.S.; investigation, T.L.N. and D.S.; resources, T.L.N. and D.S.; data curation, T.L.N. and D.S.; writing—original draft preparation, T.L.N. and D.S.; writing—review and editing, T.L.N. and D.S.; visualization, T.L.N. and D.S.; supervision, T.L.N. and D.S.; project administration, T.L.N. and D.S.; funding acquisition, T.L.N. and D.S. All authors have read and agreed to the published version of the manuscript.

**Funding:** This research was funded by Faculty of Organization and Management, Silesian University of Technology, Poland, grant number 13/010/BK_22/0065.

**Institutional Review Board Statement:** Not applicable.

**Informed Consent Statement:** Informed consent was obtained from all subjects involved in the study (as the survey was voluntary and anonymous, informed consent was obtained with its completion).

**Data Availability Statement:** Data are contained within the article.

**Conflicts of Interest:** The authors declare no conflict of interest.

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
