# Peer review of "The Importance of Selected Aspects of a Company’s Reputation for Individual Stock Market Investors—Evidence from Polish Capital Market"

_sustainability, doi:10.3390/su14159187_

Round 1
Reviewer 1 Report
allows for gaining media favor and journalists,
needs a rewrite here. Should be favor of media and journalists
well as the amount of prices,
better to write simply as well as prices
important for local communities are 86 responsibility and social commitment of the company
run on sentence
According to the results of the 94 research by Weber Shandwick [20], global executives consider the most important thing 95 Sustainability 2022, 14, x FOR PEER REVIEW 3 of 16 for the company's reputation that the perception of three groups of stakeholders: cus- 96 tomers, investors and employees (Table 1).
Bad sentence.
the company's involvement in CSR activities,
I don’t know what CSR means.
The conducted analysis of the literature allows us to answer the first research ques- 183 tion (Q1).
You need to tell us up front what the first research question is, which you have not done yet. Better than The conducted analysis is Our analysis
The main goal of the article, i.e. to examine the importance of selected aspects of 191 companies' reputation for individual investors,
Better to write The main goal of the article is to examine
well as social issues, was achieved by conducting a survey using the google- 193 docs form. Surveys allowing to achieve the main goal and answer the research questions 194 were carried out on a randomly selected representative sample of 417 individual inves- 195 tors out of 1.356 million (number of brokerage accounts in Poland in the end of November 2021), assuming a confidence level of 0.95, a maximum error of 5%, and a fraction of 0.5.
This makes no sense. I don’t know what a fraction of 0.5 means in this instance
Table 4: left hand column needs to be complete. No ….
This is an interesting paper. Investors want to be sure they are not being ripped off. Consequently, they want information that they can trust. Perhaps this would be a good last line to the article. I support publication with the few suggestions noted above. In their next paper I would like to see the authors test the hypothesis that better information leads to more success for companies.

Author Response
Dear Professor, Thank You very much for the review and Your commitment. All Your suggestions were treated as valuable and significant and they helped us to improve the scientific value of the research. A detailed response for Your comments is given below.
Kind regards,
Authors

Reviewer 2 Report
The article presents a study on the importance of reputation in the evaluation that individual investors make of companies in the Polish stock market.
It starts with an Introduction that loosely justifies the need and aims for the work (eg in lines 55/56: "Until now, many authors and researchers have discussed the importance of reputation for investors in their studies" - No reference and no quantification of "many authors and researchers" and in line 58 state that "the aim of this article was to fill this gap"). It can be improved by citing literature references regarding "the importance of reputation for investors").
The second chapter (Literature review) is composed of two sections with the same title. Please correct.
Between lines 149 and 152, there are no references, in spite of being said: "Research conducted in recent years in many countries ..." and "Based on the literature review ...". In both cases, the article can be improved by citing supporting references.
From line 183 onwards, it is stated "The conducted analysis of literature allow us to answer the first research question (Q1).". (until this point no presentation of research questions was made (it is made between line 226 and 238 ..)) and concludes from the literature review that "investing in reputable companies involves less risk". There is very little evidence in the literature to support this conclusion and no quantitative data is presented. Also, the research questions presented later between lines 226 and 238 are also very loosely supported by the literature review performed. Work can be improved if, from the literature review presented, all the research questions can be supported and if conclusions are all placed at the end of the article.
In the "Methods and materials" chapter, it is not clear how the sample of 417 respondents was selected (it appears that the respondents are surveyed in periodic surveys of the Association of Individual Investors in Poland ... line 205). Also, there is no reference to the number of investors addressed, nor the indication of the response ratio.
Regarding the questionnaire, the topics addressed are loosely justified with a set of 5 references (39, 78, 79, 80 and 24), the scope of publicly available information and the author´s experience in the stock market. It is suggested that a more systematic and thorough analysis and justification can be performed, deriving from the literature review of the research questions and the topics addressed.
Concerning the "scientific method", the research questions could have been transformed into hypotheses so that they can be validated (or not) by the data and the respondents could have been separated according to the dimensions presented in the questionnaire (eg: investment experience). Using this strategy would have allowed a much richer presentation of results and discussion than the one obtained.
In what concerns the "Results", there is a mere presentation of averages and "averages of averages" of the responses obtained that result from the approach taken in the "Methods and materials" chapter.
The "Discussion" chapter does not make sense, since it references similar studies and the results achieved in different countries. This information should have probably been placed in the "Literature review" chapter and served as the basis for the research questions
The "Conclusions" chapter presents conclusions that are loosely related to the results found. The final phrase, and as an improvement, presents one of the approaches that should have supported the approach to the present article: "... taking into account different criteria differentiating investors according to gender, age or investment experience."
Author Response

(The authors gave the same response as above.)

Round 2
Reviewer 2 Report
Still lacks important statistical analysis of the data gathered.
Chapter "6. Conclusions"
Author Response
Dear Professor,
Thank You very much for the review and Your commitment.
A detailed response for Your comment is given in the attached pdf file.
Kind regards,
Authors
